# Structure of human TRPM8 channel

Sergii Palchevskyi[1,2,6], Mariusz Czarnocki-Cieciura [1,6], Giulio Vistoli[3], Silvia Gervasoni [3,4], Elżbieta Nowak[1], Andrea R. Beccari[5], Marcin Nowotny [1✉] & Carmine Talarico [5✉]

TRPM8 is a non-selective cation channel permeable to both monovalent and divalent cations that is activated by multiple factors, such as temperature, voltage, pressure, and changes in osmolality. It is a therapeutic target for anticancer drug development, and its modulators can be utilized for several pathological conditions. Here, we present a cryo-electron microscopy structure of a human TRPM8 channel in the closed state that was solved at 2.7 Å resolution. Our structure comprises the most complete model of the N-terminal pre-melastatin homology region. We also visualized several lipids that are bound by the protein and modeled how the human channel interacts with icilin. Analyses of pore helices in available TRPM structures showed that all these structures can be grouped into different closed, desensitized and open state conformations based on the register of the pore helix S6 which positions particular amino acid residues at the channel constriction.

[1] Laboratory of Protein Structure, International Institute of Molecular and Cell Biology in Warsaw, 02-109 Warsaw, Poland. [2] Cell Signalling Department, Institute of Molecular Biology and Genetics NASU, 03143 Kyiv, Ukraine. [3] Dipartimento di Scienze Farmaceutiche, Università degli Studi di Milano, Via Mangiagalli, 25, I-20133 Milano, Italy. [4] Department of Physics, University of Cagliari, I-09042 Monserrato, Italy. [5] Dompé Farmaceutici SpA, EXSCALATE, Via Tommaso De Amicis, 95, I-80131 Napoli, Italy. [6] These authors contributed equally: Sergii Palchevskyi, Mariusz Czarnocki-Cieciura.
✉email: mnowotny@iimcb.gov.pl; carmine.talarico@dompe.com

TRPM8 is a non-selective $Ca^{2+}$-permeable transient receptor potential (TRP) channel. It belongs to the TRPM (melastatin) family that comprises eight members (TRPM1-8) that are involved in processing different stimuli, including temperature, taste, pressure, changes in osmolality, and oxidative stress[1]. TRPM8 is a multimodal sensor of innocuous-to-noxious cold that is activated by several factors, such as temperature, cooling agents (e.g., menthol and icilin), voltage, pressure, and osmolality[2]. It is mainly expressed in sensitive primary afferent neurons that innervate cold-sensitive tissues, such as the skin, teeth, nasal mucosa, and tongue[3]. TRPM8 has attracted great interest because of the therapeutic applications of its modulators that can be utilized for several pathological conditions, including neuropathic pain, irritable bowel syndrome, oropharyngeal dysphagia, chronic cough, and hypertension[4]. Importantly, TRPM8 is involved in various processes that are related to cancer progression, suggesting that its ligands can exert anticancer activity in tissues that express TRPM8, such as breast cancer, bladder cancer, esophageal cancer, lung cancer, skin cancer, pancreatic cancer, colon cancer, gastric cancer, and osteosarcoma[5]. Because of its importance, a significant number of TRPM8 modulators have been reported[6]. Nevertheless, to support further structure-based rational drug design, the binding sites of this ion channel and the mechanism of its modulation by ligands, particularly for the human protein, need to be better explored[7].

The first TRPM8 structures were solved by cryo-electron microscopy (EM) for proteins from birds (i.e., *Ficedula albicollis*, FaTRPM8[8,9] and *Parus major*, PmTRPM8[10]). Both are highly homologous with the human TRPM8 channel, with a sequence identity of 83% and 80%, respectively. The reported structures correspond to the apo and agonist- or antagonist-bound states. Notably, all avian structures are in the closed state, and binding of the agonist (icilin, WS12)[9] or antagonist (AMTB, TC-I 2014)[10] introduces only minor structural rearrangements. A much larger difference was observed for the structure that was solved in the presence of $Ca^{2+}$ ions, which was defined as a desensitized (DS) state[10]. All avian TRPM8 structures show significant differences compared with the available mammalian structures of other TRPM channels (i.e., for TRPM4, TRPM5, and TRPM7), including conformation of the S4-S5 linker that connects transmembrane helices S4 and S5 in the apo state and the overall S5-S6 arrangement in $Ca^{2+}$-bound structures[11]. Finally, a set of mammalian TRPM8 structures from mouse (MmTRPM8) showed rearrangements of the transmembrane region of the TRPM8 channel in closed ($C_0$, $C_1$), intermediate ($C_2$), and open (O) states[12]. These structures revealed that the binding of the membrane signaling lipid phosphatidylinositol-4,5-bisphosphate ($PIP_2$) is required for the activation of mouse TRPM8 channels. Additional MmTRPM8 structures were solved in apo form, in a $Ca^{2+}$-bound state, and in complex with icilin and $Ca^{2+}$ ions[13]. They all shared the same closed state that is very similar to the DS state of PmTRPM8[10] (with a canonical S4-S5 linker that resembles other mammalian TRPM proteins), presumably due to the $PIP_2$ depletion and additional interactions with CHS molecule[12]. Despite remarkable advancements in understanding TRPM8 that have been conferred by these avian and mouse structures that allowed TRPM8 desensitization, activation, and opening mechanisms to be proposed, several questions remain unanswered. Additional mammalian structures will further clarify the observed structural features and help interpret reported differences among TRP channels. Furthermore, no direct structural information is currently available for the human TRPM8 channel, and the N-terminal region that contains the first two melastatin homology regions (MHR) and pre-MHR domain is poorly resolved in all available structures.

To better understand the structural features of the TRPM8 ion channel, we solved a cryo-EM structure of the human TRPM8 protein in $C_0$ state at 2.7 Å resolution. By combining a high-resolution global map with focused refinement maps of N-terminal domains, we were able to build an atomic model of human TRPM8 (HsTRPM8) that comprises residues 43–1104.

## Results

**Cryo-EM structure of the human TRPM8 channel**. We determined a cryo-EM structure of apo HsTRPM8 that was solubilized with lauryl maltose neopentyl glycol (LMNG) and cholesteryl hemisuccinate (CHS). It corresponds to a closed $C_0$ state of the pore[12] (Fig. 1a, b, Supplementary Figs. 1 and 2, Table 1), as classified based on the arrangement of S5 and S6 helices (see Discussion). At 2.7 Å resolution, it is one of the two highest-resolution structures of any TRPM8 protein that is currently available. Overall, human TRPM8 adopts a characteristic homo-tetrameric arrangement and is very similar to previously described TRPM8 structures[8–10,12,13]. The folded N-terminal part of the protein (residues ~43–100), termed the pre-MHR, is followed by four MHRs (MHR1-4; Fig. 1c, d). Together they comprise ~630 residues and form the majority of the cytoplasmic part of the channel[14,15]. Melastatin homology regions are followed by a transmembrane domain (TMD) with six transmembrane helices characteristic of many ion channels, including all proteins from the TRP superfamily. It resembles TMDs that are present in TRPV1[16] and TRPV2[17] and is composed of a pre-S1 domain, a voltage-sensor-like domain (VSLD), and the pore. The TMD is followed by the TRP helix that is involved in the formation of the binding pocket for cooling agents[9,10,12,13]. The C-terminal domain (CTD) is composed of two helices (CTDH1/2) and a long coiled-coil element that extends toward the cytosolic part of the structure, and is important for channel assembly and function[15,18].

**The pore of HsTRPM8 is in a closed $C_0$ state**. The transmembrane region of TRPM8 adopts a fold characteristic of proteins from the TRP family, with helices S5-S6 swapped between adjacent subunits of the tetramer. The pore is composed of helices S5 and S6 connected with a pore helix (PH) (Fig. 2a). The lower gate of the channel is formed by hydrophobic side chains of M978 and F979 (both shown in Fig. 2a as sticks) that close the pore. The upper gate is formed by the selectivity filter and is followed by the outer pore loop (residues 914–951) which links the PH and S6 helices. This loop is not fully resolved in our reconstruction. An extensive structural comparison of all TRP channels[19] suggested that selectivity filters of some non-selective TRP channels are intrinsically flexible. We speculate that the filter is stabilized in specific conformations (e.g., in DS state), thus allowing its visualization in the other cryo-EM structures.

The conformation of the pore in our structure is similar to the closed $C_0$ MmTRPM8[12]. However, it is different from the conformation that was observed in the MmTRPM8 structures that were solved by using comparable CHS concentrations[13] in which the S6 helix is rotated, positioning a different residue (V976) to close the lower gate (Fig. 2b). This rearrangement is accompanied by stabilization of the upper gate and selectivity filter which are visible in these $PIP_2$-depleted DS MmTRPM8 structures and in the DS PmTRPM8 structure[10]. In contrast, in our reconstruction, only the first two amino acids from the selectivity filter (F912 and G913) are resolved. Important differences are also observed in the VSLD, composed of helices S1-S4. In the DS structures, helices S1-S4 are shifted, and the conserved TRP helix is positioned more parallel to the

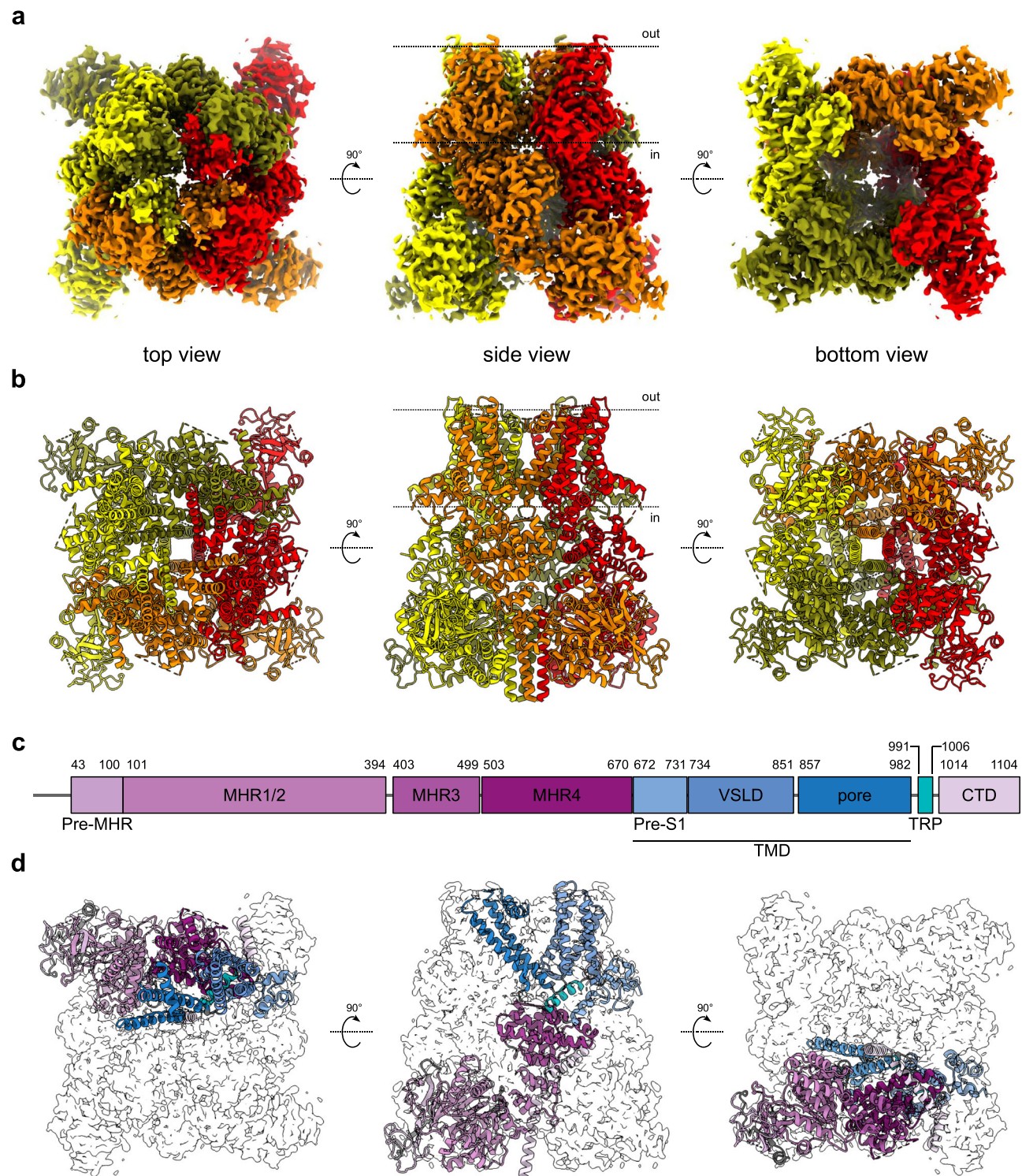

**Fig. 1 Overall structure and domain composition of human TRPM8 channel. a** Cryo-electron microscopy electrostatic potential composite HsTRPM8 map colored according to the channel subunits (olive, red, orange, and yellow; three views). **b** Atomic model of HsTRPM8 in cartoon representation, colored as in **a**. **c** Domain composition of HsTRPM8. Residue numbers at the boundaries between particular elements are given. **d** Structure of a single subunit with domains colored as in **c**. A contour of the cryo-EM composite map for the entire tetramer is shown in gray (three views). Dotted lines in **a**, **b** indicate the approximate boundaries of the cell membrane. MHR melastatin homology region, VSLD voltage-sensor-like domain, TMD transmembrane domain, TRP TRP helix, CTD C-terminal domain. See also Supplementary Figs. 1 and 2.

membrane bilayer (Fig. 2c, d). Finally, substantial differences are observed in the S4 and S5 helices (Fig. 2c, e). In DS MmTRPM8, a canonical S4-S5 linker is formed, and the S4 helix adopts a $3_{10}$ helical conformation at residues 841-LRL-843. In our structure,

the S4-S5 linker is parallel to the S5 helix, and the S4 helix is fully α-helical. We performed a similar comparison for other published structures of TRPM channels (see Supplementary Figs. 3 and 4, and Discussion for details).

**Table 1 Cryo-EM data collection, refinement, and validation statistics.**

|  | Composite TRPM8 map (EMDB-15981) (PDB 8BDC) | Consensus TRPM8 map (EMDB-15982) | Focused MHR1/2 map (EMDB-15983) |
|---|---|---|---|
| Data collection and processing |  |  |  |
| Magnification | 105,000× | 105,000× | 105,000× |
| Voltage (kV) | 300 | 300 | 300 |
| Electron exposure (e–/Å$^2$) | 41.42 | 41.42 | 41.42 |
| Defocus range (µm) | −0.9 to −2.7 | −0.9 to −2.7 | −0.9 to −2.7 |
| Pixel size (Å) | 0.82 | 0.82 | 0.82 |
| Symmetry imposed | C4 | C4 | C1 |
| Initial particle images (no.) | 1,239,855 | 1,239,855 | 1,239,855 |
| Final particle images (no.) |  | 110,176 | 212,851 |
| Map resolution (Å) |  | 2.65 | 3.20 |
|  FSC threshold: |  | 0.143 | 0.143 |
| Map resolution range (Å) |  | 2.36–35.25 | 2.88–12.10 |
| Refinement |  |  |  |
| Initial model used (PDB code) | 6O6A, 6O72, 7WRA |  |  |
| Model resolution (Å) | 2.70 |  |  |
|  FSC threshold | 0.5 |  |  |
| Model resolution range (Å)* | 2.38–5.43 |  |  |
| Map sharpening $B$ factor (Å$^2$) | −87 |  |  |
| Model composition |  |  |  |
|  Non-hydrogen atoms | 30,132 |  |  |
|  Protein residues | 3852 |  |  |
|  Ligands | 24 |  |  |
| $B$ factors (Å$^2$) |  |  |  |
|  Protein | 55.04 |  |  |
|  Ligand | 42.92 |  |  |
| R.m.s. deviations |  |  |  |
|  Bond lengths (Å) | 0.002 |  |  |
|  Bond angles (°) | 0.394 |  |  |
| Validation |  |  |  |
|  MolProbity score | 1.68 |  |  |
|  Clashscore | 5.33 |  |  |
|  Poor rotamers (%) | 2.19 |  |  |
| Ramachandran plot |  |  |  |
|  Favored (%) | 97.25 |  |  |
|  Allowed (%) | 2.75 |  |  |
|  Disallowed (%) | 0.00 |  |  |

*Range of the local resolution map values at atom positions.

**Analysis of ligand-binding pockets in TMD**. The main ligand-binding pocket in TRPM8 proteins, termed the VSLD cavity, is formed by residues from transmembrane helices S1-S4 and TRP helix[8]. Previous studies indicated that residues R842 and K856 contribute to the voltage dependence of TRPM8[20], whereas R842, Y745, and Y1005 have been shown to interact with menthol[20,21]. This is in agreement with the cryo-EM structure of PmTRPM8 and FaTRPM8, which visualized several ligands (including antagonists) in this pocket[9,10,13]. We collected cryo-EM data for HsTRPM8 in complex with several ligands, but we could not observe a clear density in this cavity (data not shown). However, in our apo HsTRPM8 reconstruction, we observed additional densities that clustered in putative ligand-binding pockets around the TMD (Fig. 3a, blue). In agreement with previous reports[10], we observed a discrete density in the VSLD cavity that partially overlaps with icilin in the FaTRPM8 structure[9] (Fig. 3b). Additional well-defined densities are also visible inside the channel between helices S5 and S6. We and others[10] assigned these to an unidentified phospholipid tail (modeled as undecane) and two Na$^+$ ions (Fig. 3c). Finally, in the vicinity of the swapped helix S5 from one tetramer subunit and VSLD of the adjacent subunit, we observed clear densities that were assigned previously to CHS, 3-SN-phosphatidylethanolamine (9PE), and another undecane molecule[10]. The quality of our EM map in this region allowed us to unambiguously assign them to two CHS molecules and a phospholipid with one monounsaturated fatty acid which we modeled as phosphatidylcholine (POPC; Fig. 3d). In summary, additional densities visible in our HsTRPM8 (and other published structures) can be interpreted as compounds copurifying with the protein (PE/POPC, Na$^+$) or used for its solubilization (CHS), but we cannot be sure about their identity and source, and thus interpret their physiological relevance. Importantly, we did not observe any density in the putative Ca$^{2+}$-binding pocket between the S2 and S3 helices that could be assigned to the divalent ion, which further confirms that our structure is not in the desensitized state.

Previous studies showed that binding to ligands does not significantly affect the conformation of the pore, so we could use our structure to model a HsTRPM8 complex with icilin. By considering the two different poses that are assumed by icilin in the resolved structures[9,13], two models were generated. The first model was obtained by superimposing the sensor module (S1-S4) of HsTRPM8 with avian FaTRPM8 in complex with icilin (PDB ID: 6NR3). The second model was obtained by manually rotating the icilin from the first complex by 180° to mimic the pose that is observed in the DS MmTRPM8 structures[13]. The comparison of the complexes that were obtained for the four monomers of the HsTRPM8 tetramer in the first model reveals two different binding

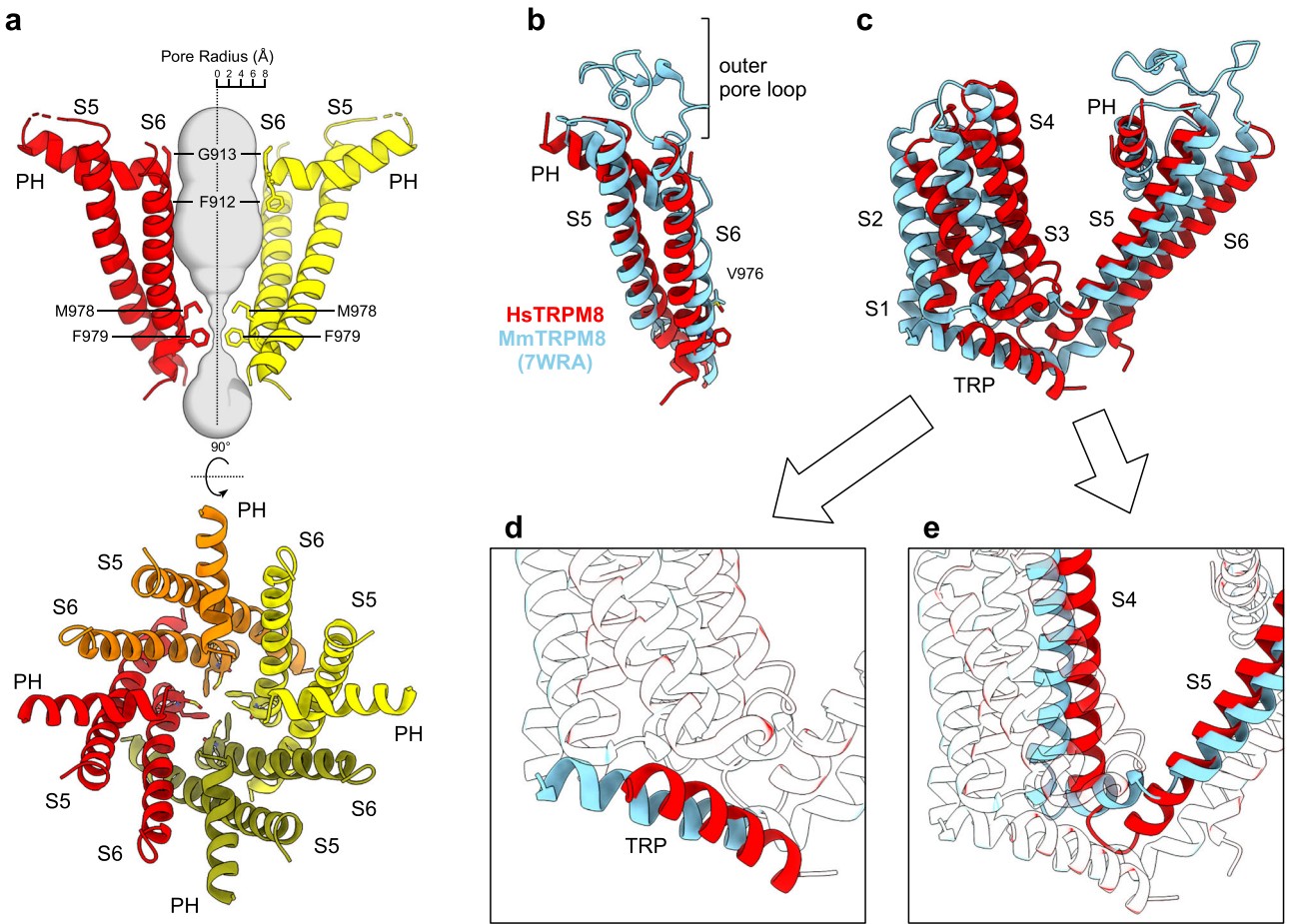

**Fig. 2 Configuration of the HsTRPM8 TMD in a closed $C_O$ state. a** The pore of HsTRPM8, composed of helices S5, PH, and S6 (two views). Hydrophobic residues M978 and F979 closing the gate and the F912 and G913 from the selectivity filter are shown as sticks. In the upper panel, front and rear subunits were removed for clarity. Gray surface represents pore radius calculated using the HOLE program[43]. **b** Comparison of the pore of HsTRPM8 (red) and MmTRPM8 (light blue) (PDB ID: 7WRA). **c–e** Comparison of TMDs of HsTRPM8 (red) and MmTRPM8 (light blue) after superimposition of the corresponding pore regions. Insets show details of differences in the conformation of TRP helix (**d**) and S4-S5 linker (**e**). PH pore helix, TRP TRP helix. See also Supplementary Figs. 3 and 4.

modes that mostly differ in the arrangement of the phenoxy ring (Fig. 3e, f). Indeed, in all monomers, the dihydropyrimidinone ring approaches E782 and R842, and the nitrophenyl moiety interacts with N741 and Y1005. In contrast, only in one binding mode (represented by monomer A), the phenoxy group interact with Y745. In the other mode (monomer C), it elicits clear π-π stacking with F839, whereas Y745 approaches the dihydropyrimidinone ring. The analysis of a set of representative scoring functions (Supplementary Table 1) suggests that the complex of monomer C is more stable than the complex of monomer A, likely because of stronger hydrophobic contacts.

The analysis of rotated poses also revealed differences between the four subunits, mainly involving interactions that were formed by the E782/R842 dyad (Fig. 3g, h). They point toward the nitrophenyl ring in monomer A. In monomer B, they assume a more central arrangement by which they contact all three rings. In both monomers, Y745 approaches the dihydropyrimidinone ring, whereas the phenoxy moiety contacts N741 and Y1005. The analysis of corresponding scores revealed that the complex of monomer B is more stable, mostly because of stronger polar interactions. Finally, the comparison of docking scores as computed for the original and rotated poses did not reveal significant differences, suggesting that both poses are similarly plausible within the explored binding site (Supplementary Table 1).

To validate our modeling of icilin binding, we generated a set of variants of HsTRPM8 protein in which selected residues in the putative ligand-binding pocket (Y745, I746, N799, and D802) were mutated to alanine. Next, we tested the activation of wild-type HsTRPM8 and its point mutants in transiently transfected HEK293 cells by icilin and menthol-related molecules (menthol, Cooling Agent-10, and WS-3). As a readout, we used a normalized fluorescence signal of $Ca^{2+}$ mobilization-dependent dye, Fluo-8 NW (Fig. 3i, Supplementary Fig. 5). Y745 and D802 are located in S1 and S3 helix, respectively, and are directly interacting with icilin in our models. I746 and N799 are located in the vicinity of these residues but are not predicted to be involved in ligand binding, so we used them as negative controls. For wild-type protein, the $EC_{50}$ values for icilin, Cooling Agent-10, and WS-3 were similar to the values reported in previous studies[22]. $EC_{50}$ values reported for menthol varied significantly between studies[22,23] and value obtained in our assay lies in the same range. In agreement with the modeling, Y745A variant completely lost its activatability for all tested ligands, and D802 variant showed much weaker response. In contrast, I746A mutation caused only moderate effect. N799A variant showed strong inhibition of icilin binding but interactions with other ligands were not affected. This unexpected effect notwithstanding, these results confirm the importance of selected residues for icilin binding and support the modeled binding mode.

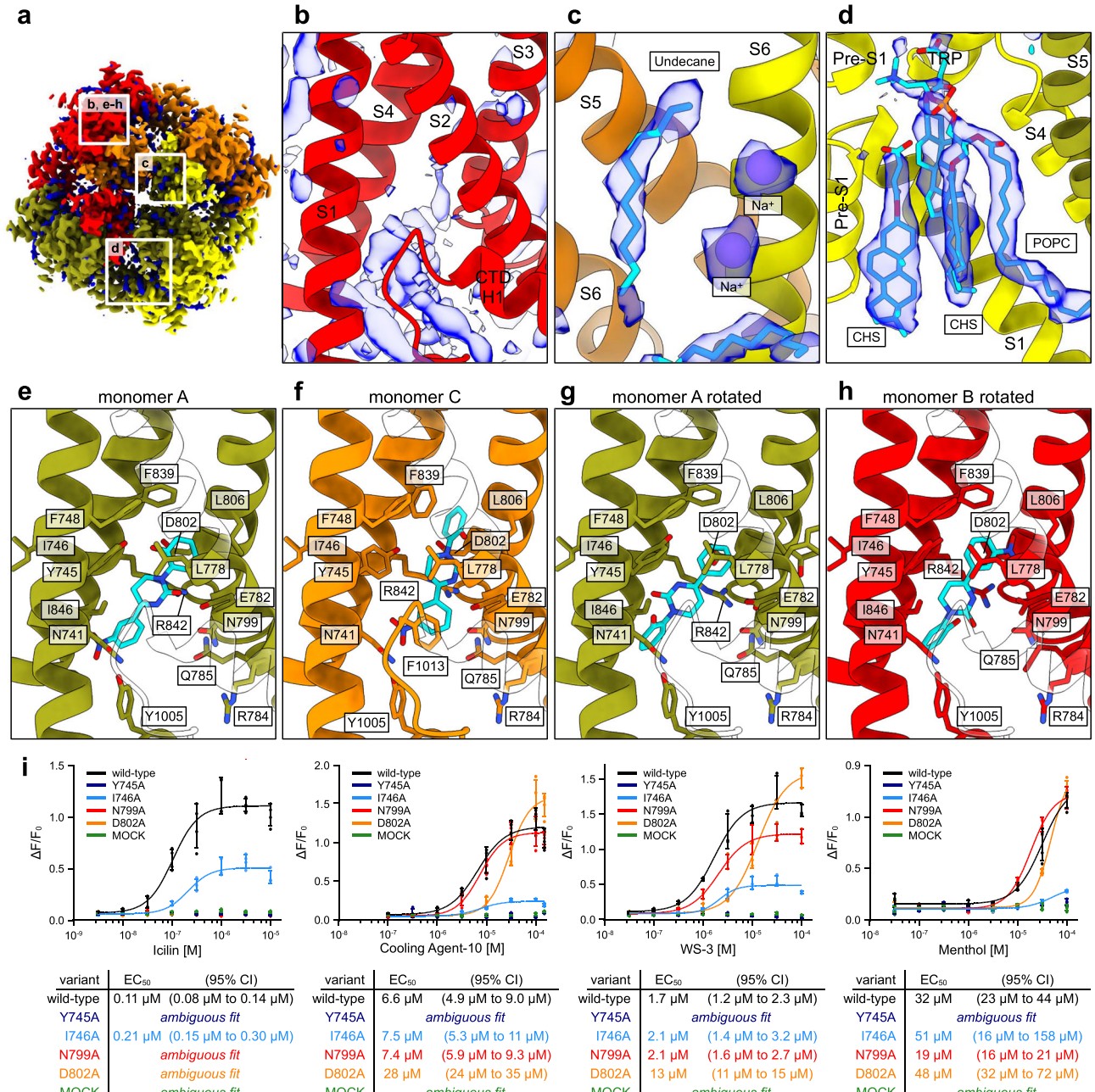

**Fig. 3 Ligand-binding pockets in HsTRPM8, modeling, and validation of icilin binding. a** HsTRPM8 composite map (top view) with non-protein densities colored blue. **b** VLSD cavity with unidentified non-protein densities. **c** Densities between helices S5 and S6 are assigned to undecane and $Na^+$ ions. **d** Densities between helix S5 and VSLD assigned to two molecules of cholesteryl hemisuccinate (CHS) and one phosphatidylcholine (POPC). **e**–**h** Modeling of icilin bound to VSLD cavity of HsTRPM8 in standard (**e**, **f**) and rotated (**g**, **h**) orientation. Predicted interacting residues are labeled and displayed as sticks. **i** HsTRPM8 activation assay: concentration-response curves of HsTRPM8 variants activated by selected agonists. HEK293 cells were transiently transfected with constructs coding indicated HsTRPM8 variants or empty vectors (MOCK). Transfected cells were then stimulated with increasing concentrations of the indicated agonists. Dose response curve with a variable slope was fitted to a measured fluorescence signal of the Fluo-8 NW dye normalized to the cellular responses prior to agonist stimulation ($\Delta F/F_0$). Error bars correspond to SD from $n = 3$ (WS-3) or $n = 4$ (Icilin, Cooling Agent-10, Menthol) replicates. For each curve, $EC_{50}$ value is given with 95% confidence intervals. See also Supplementary Figs. 2e and 5, and Supplementary Table 1.

**Structure of the Pre-MHR and MHR1/2 domains**. In our initial reconstruction, fragments of the N-terminal region of the protein, spanning amino acids 1–400 (pre-MHR and MHR1/2 domains), were not clearly defined. Three-dimensional (3D) variability analysis in cryoSPARC suggested that this region is quite mobile, and its movements correlate with minor rearrangements of the whole pore structure (Supplementary Movie 1). Interestingly, some of these movements appear to preserve two-fold symmetry

rather than four-fold symmetry of the channel, which is in agreement with the reported TRPM2 structures that showed intermediate states with C2 symmetry[9]. These regions were also not well resolved in the other reported TRPM8 structures[8–10,13]. To improve the quality of the reconstruction of this region we performed its focused refinement, followed by C4 symmetry expansion and subtraction of the signal from the rest of the TRPM8 molecule. This resulted in 3.2 Å reconstruction with

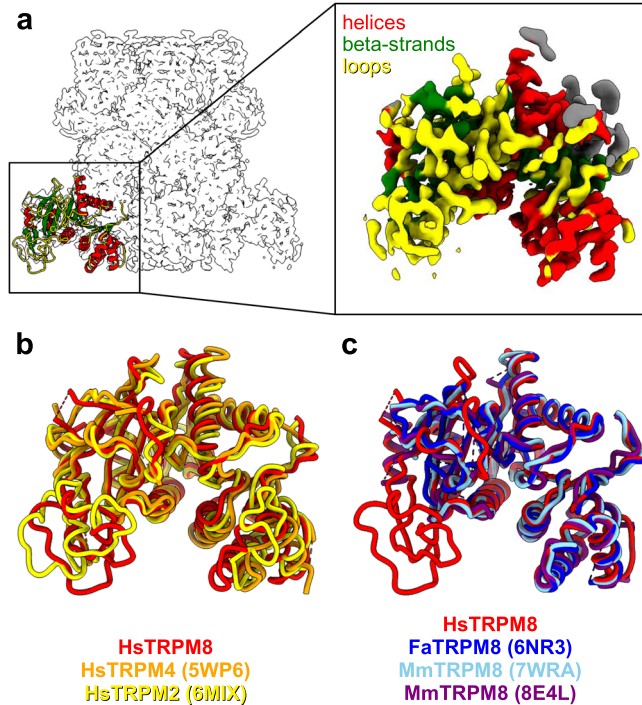

**Fig. 4 Structure of the pre-MHR and MHR1/2 domains of HsTRPM8.**
**a** Atomic structure of the pre-MHR and MHR1/2 from one subunit of the HsTRPM8 is shown in cartoon representation with helices in red, α-strands in green, and loops in yellow. A contour of the composite cryo-EM map for the entire tetramer is shown in gray. The inset shows a focused MHR1/2 map in the same color code. **b**, **c** Comparison of pre-MHR from HsTRPM8 reported herein (red) with other human TRPM channels (**b**) and orthologous TRPM8 structures (**c**) (colored as in the figure keys).

quality that was sufficient for the de novo building of amino acids 43–140 of the protein chain that were not clearly visible in all previous reconstructions (Fig. 4a).

The DALI search that was performed for this region showed that it adopts a canonical SLOG-like fold that is found in N-terminal regions of many other channels from the TRPM family, despite very low sequence identity (Fig. 4b, <30% identity for the shown structures). Interestingly, this region is involved in ligand binding in some TRP channels[24,25], suggesting its role in the sensing of external stimuli. In TRPM4, it is referred to as the N-terminal nucleotide-binding domain (NBD) and is responsible for binding adenosine triphosphate[24]. It also contains an additional binding pocket for the TRPM4 modulator decavana-date (DVT)[25]. The NBD is highly similar to the MHR1/2 domain in TRPM8, but the residues that are responsible for nucleotide-binding and the overall topology of the DVT-binding pocket are not conserved. We note that in our focused map, this region is better resolved (including a long loop that comprises residues 55–94) than in any other available TRPM8 structure (Fig. 4c). Our model thus provides the most complete visualization of N-terminal portion of the TRPM8 channel to date.

## Discussion

Recent years have brought several cryo-EM structures of TRPM8 channels[8–10,12,13]. Yin and colleagues classified some of them based on the arrangement of the pore region into closed ($C_0$, $C_1$), intermediate ($C_2$), and open (O) states[12]. Another nonconducting state (DS) was observed after calcium-dependent desensitization of the channel[10].

To put our reconstruction into the context of available structural data, we compared the pore region in our HsTRPM8 with corresponding parts of other available structures of TRPM8 channels. Conformation of the pore was the same between the human structure and the closed $C_0$ state of the mouse (MmTRPM8, PDB ID: 8E4P; Supplementary Fig. 3a) and avian (FaTRPM8, PDB ID: 8E4Q; Supplementary Fig. 4b, $C_0$ state) TRPM8 channels[12]. The same conformation was also observed for most avian structures, including PmTRPM8 in apo form (PDB ID: 6O6A) and in complex with the antagonists TC-I 2014 (PDB ID: 6O72) and AMTB (PDB ID: 6O6R), and for FaTRPM8 in the apo state (PDB ID: 6BPQ) and in complex with the icilin analog WS12 (PDB ID: 6NR2). Notably, a different pore con-formation was observed for the $Ca^{2+}$-bound DS structure of PmTRPM8[10] (PDB ID: 6O77) and for the $PIP_2$-depleted DS MmTRPM8 structures[13] (PDB ID: 7WRA-F; Supplementary Fig. 3b). We performed a similar comparison for pore regions of other members of the TRPM family. Pore helices from MmTRPM4[24] (PDB ID: 6BCJ) and MmTRPM7[26] (PDB ID: 5ZX5) superimpose well with their counterparts in the $PIP_2$-depleted DS MmTRPM8 structures[13] (Supplementary Fig. 3c). This suggests that the reported structures of MmTRPM4 and MmTRPM7 channels are in a conformation that is similar to the desensitized state of PmTRPM8, which is different from the $C_0$ state observed in our HsTRPM8 structure and abovementioned mouse and avian TRPM8 structures.

TRP channels can also be classified based on the configuration of the S6 helix. The structural alignment of all available TRP channel structures[19] revealed that the lower gate can be con-stricted at four sites (A, B, C, and D), spanning three helical turns of the S6 helix (Supplementary Fig. 4a). In most TRP structures, the pore is closed by a small and hydrophobic residue at site B (L/ I/V), accompanied by a less conserved polar or hydrophobic residue from site C. All reported $PIP_2$-depleted DS MmTRPM8 structures (PDB ID: 7WRA-F) and the $Ca^{2+}$-desensitized PmTRPM8 channel (PDB ID: 6O77) have the same S6 helix configuration (referred to as "0" register shift). In con-trast, the conformation of other available TRMP8 structures in closed $C_0$ state (including the one reported herein) is different, with the register of S6 helix altered by half a turn. This forces different residues ("+2" according to the above register or "−2" for the FaTRPM8 protein that was solved with $PIP_2$ and icilin at high occupancy [PDB ID: 6NR3]) to constrict the channel. In turn, in human and most avian structures, the pore is closed through hydrophobic interactions of side chains of F979 and M978 (site B), followed by T982 (site C). In the C1/C2 states (PDB ID: 8E4M, 8E4N, 8E4O), upon $PIP_2$ binding S6 helix in MmTRPM8 channel is even more rotated ("+3" register shift), exposing negatively charged or polar residues to the center of the pore. Finally, the MmTRPM8 in open state (PDB ID: 8E4L) has the same register ("0") as the DS structures. Overall, these dif-ferences in configuration of the S6 helix show the flexibility of the lower gate and how it can respond to different stimuli.

The register shift in the S6 helix from closed $C_0$ ("+2) to DS ("0") state is accompanied by a number of conformational changes in the pore region[2,10]. These include (i) a rigid-body tilt of the VSLD, (ii) the formation of a canonical S4-S5 linker, (iii) shifts of the S5, PH, and S6 helices, (iv) stabilization of the outer pore loop, (v) tilting of the TRP domain, and (vi) the introduction of a $3_{10}$-helix in S4 and π-helix in both the PH and S6 helices. Our analyses imply that our HsTRPM8 structure resembles the $C_0$ closed state that was described for MmTRPM8 protein[12] and is shared with most of the avian channels, whereas the $PIP_2$-depleted MmTRPM8 structures[13] are very similar to the desen-sitized state described for PmTRPM8[10].

The structure of human protein presented in this manuscript will facilitate more accurate modeling of both the influence and the binding site of various channel modulators, as well as possible conformational changes in the channel structure caused by mutations. In addition, the detailed structure of the cytoplasmic part of the channel, including pre-MHR regions, opens up additional opportunities for studying its interactions with endogenous protein modulators. The overall structure of TRPM8 channel seems to be strongly conserved among higher eukaryotes. There are only minor, mostly conservative substitutions between mouse and human sequences; most of them are located in the MHR regions and in loops that are distant from ligand/lipid binding sites. More differences can be observed between mammalian and avian TRPM8 sequences. Again, most of these substitutions are conservative and located in external loops. However, a number of amino acid replacements can be found in pre-S1 domain and S1 helix. Some of them (L697M, I701F, and V743I) are located at the interface between the protein and CHS molecules, but they are all conservative and should not have much impact on the lipid binding. Finally, over ten substitutions are located in the outer pore loop, including three between human and mouse sequences (G921S, A927S, and T932S). Unfortunately, in our HsTRPM8 reconstruction the outer pore loop is disordered, which precludes interspecies analysis of this region. Further structural studies are required to fully describe the HsTRPM8 gating mechanism and the differences between Mm and Hs proteins.

## Methods

**Reagents and chemicals.** pFastBac Dual His6 MBP N10 TEV LIC cloning vector (5 C) was purchased from Addgene (Addgene plasmid no. 30123; https://www.addgene.org/30123). All restriction enzymes, the pFastBac1 vector, and DH10Bac cells were purchased from Thermo Fisher Scientific. Sf9 cells were purchased from ATCC. Q5® High-Fidelity DNA Polymerase, the Monarch® DNA Gel Extraction Kit, and amylose resin were purchased from NEB (https://international.neb.com/). The DNA purification kit was purchased from Promega. The anti-6xHis antibodies were purchased from Abcam. All other chemicals and materials were purchased from Bio-Rad Laboratories, GE Healthcare, Sigma-Aldrich, or Roche Diagnostics, unless otherwise indicated.

**Plasmid construction, cloning, expression, purification of hsTRPM8.** Synthetic cDNA that encoded full-length *Homo sapiens* TRPM8 (HsTRPM8, NCBI Reference Sequence NM_024080.5) was codon optimized for *S. frugiperda* and cloned into a pEX-K248 vector by Eurofins Genomics (https://eurofinsgenomics.eu/). cDNAs of 6xHis-MBP-TEV from the pFastBac Dual His6 MBP N10 TEV LIC cloning vector (5 C) and hsTRPM8 from the pEX-K248 vector were polymerase chain reaction (PCR) amplified with Q5® High-Fidelity DNA Polymerase, gel purified by the Monarch® DNA Gel Extraction Kit, and subcloned into a pBastBac1 expression vector using the EcoRI restriction site. The final His-MBP-TEV-hsTRPM8 construct contained full-length unmodified sequences of human TRPM8. This construct was used for DH10Bac *E. coli* cell transformation with subsequent bacmid isolation by isopropanol precipitation from appropriately selected colonies. To prepare baculovirus, Sf9 insect cells were transfected by PCR-verified His-MBP-TEV-hsTRPM8-containing bacmids in a six-well plate according to the manufacturer's protocol (Bac-to-Bac, Invitrogen). After ~6–7 days, the low-volume P1 generation virus that contained cell media was harvested, spun down, and stored at 4 °C. The P1 generation cell medium was used to create the high-volume/high-titer P2 generation virus stock. To prepare P2, 25 ml of cells at a density of ~2 million cells/ml were infected by P1 media at a 1:100–200 ratio and incubated for 6–7 days, harvested, spun down, stored at 4 °C, and used to prepare large-scale His-MBP-TEV-hsTRPM8 expression.

Sf9 cells between the 5th and 20th passage were utilized for His-MBP-TEV-hsTRPM8 expression. Briefly, cells were kept in ESF 921 Insect Cell Culture Medium (Expression systems) that was supplemented with 10% (v/v) fetal bovine serum (Gibco) and 1% Antibiotic Antimycotic Solution (Gibco) in an orbital shaker incubator that was set at 130 rotations per minute (rpm) and 27 °C. They were cultured in a sterile 2 L Erlenmeyer flask in a 400–500 ml volume to density ~1.5 million cells/ml. The culture was then infected by the addition of the P2 generation virus stock at a final ratio of 1:100–200 (v/v), incubated for 72 h, harvested, and frozen in liquid nitrogen. The pellet was stored at −80 °C. The level of His-MBP-TEV-hsTRPM8 expression was checked by anti-6xHis antibodies.

The purification procedure was performed based on a previous report[10] with some modifications. All steps were performed at 4 °C. The Sf9 cell pellet from the 400–500 ml culture was suspended in 15 ml of ice-cold buffer (20 mM HEPES [pH 7.4], 150 mM NaCl, and 2 mM Na-EGTA [pH 7.5]) that was supplemented with a cOmplete Protease Inhibitor Cocktail Tablet and Phosphatase Inhibitor Tablet (Roche) and 1 mM phenylmethylsulfonyl fluoride. To solubilize and extract hsTRPM8, 5–6 mM LMNG (Anatrace) and 1–1.2 mM CHS Tris Salt (Anatrace) were added directly to the cell lysate, and the solution was stirred for 2 h. The insoluble material was removed by centrifugation at 30,000 rpm for 50 min at 4 °C. Amylose resin (0.7–0.8 ml) was prepared by several washes with water and a final wash with buffer and incubated with His-MBP-TEV-hsTRPM8-containing supernatant for 2–3 h. Amylose resin was transferred to a gravity-flow column and sequentially washed 5–6 times with a buffer (20 mM HEPES [pH 7.4], 150 mM NaCl, 2 mM Na-EGTA [pH 7.5], 0.15 mM LMNG, and 0.03 mM CHS). The protein was then eluted in the same buffer that was supplemented with 20 mM maltose (Sigma-Aldrich). The fractions were pooled and concentrated on a Millipore concentrator (WMCO 50 kDa), and the His-MBP-tag was cleaved by TEV protease for 16–24 h at 4 °C (at a ratio of 1:10 [w/w] for TEV protease and the protein sample, respectively). Next, the TRPM8 sample was loaded onto a Superose 6 Increase size-exclusion column (GE Healthcare) that was equilibrated with SEC buffer (20 mM HEPES [pH 7.4], 150 mM NaCl, 2 mM Na-EGTA [pH 7.5], 0.025 mM LMNG, 0.005 mM CHS, and 100 μM Tris[2-carboxyethyl]phosphine [TCEP]). Fractions that corresponded to the HsTRPM8 tetramers were checked by negative staining, pooled, and concentrated to ~0.5 mg/ml using an Amicon centrifugal filter device (WMCO 100 kDa, Millipore) for cryo-EM analysis. We note that our structure was obtained using a protein that was solubilized by LMNG and CHS. Structures of other TRPM proteins that were determined using different methods (e.g., crystal *vs.* cryo-EM or detergent/amphipol solubilization *vs.* nanodisc embedding) were essentially the same[19,27]. Therefore, the method of structure determination does not appear to affect the structure of these channels.

**Cryo-EM sample preparation and data collection.** The HsTRPM8 tetramers that were solubilized by LMNG and CHS and concentrated to ~0.5 mg/ml were applied to an UltrAuFoil 1.2/1.3 mesh 300 cryo-EM grid (Jena Bioscience, catalog no. X-201-Au300), glow-discharged from both sides. The sample was vitrified in liquid ethane using an FEI Vitrobot Mark IV (Thermo Fisher Scientific) at 4 °C and 95% humidity with a 4 s blot time

and 0 blot force. Data collection was performed with a Titan Krios G3i electron microscope (Thermo Fisher Scientific) that operated at 300 kV and was equipped with a BioQuantum energy filter (with 20 eV energy slit) and K3 camera (Gatan) at the SOLARIS National Synchrotron Radiation Centre (Krakow, Poland). Movies were collected using EPU version 2.10.0.5REL with Aberration-free image shift (AFIS), with a nominal magnification of ×105,000 (corresponding to a physical pixel size of 0.82 Å), 50 μm C2 aperture, and retracted objective aperture. The defocus range was set to −0.9 to −2.7 μm. The total dose (fractionated into 40 frames) was 41.42 e/Å$^2$. The dose rate was 16.67 e/pixel/s, measured without a sample (in vacuum).

**Cryo-EM data processing**. A total of 7918 movies were collected and processed with RELION-3.1[28] and cryoSPARC v3.3.2[29] (Supplementary Fig. 1). Raw movies were motion-corrected and binned twice using RELION's implementation of MotionCor2 software[30], and 1090 micrographs with the total calculated motion larger than 100 pixels were discarded. The contrast transfer function (CTF) was fitted with CTFFIND-4.1[31], and 5,714 micrographs with the maximum CTF resolution below 5 Å were selected for subsequent processing. A total of 1.24 million particles were picked with crYOLO v1.7.4[32] and extracted with a binned pixel size of 3.28 Å. After two rounds of reference-free two-dimensional classification in cryoSPARC, 321,120 selected particles were re-imported into RELION with scripts from the University of California, San Francisco (UCSF) *pyem* package[33] and re-extracted with a pixel size of 1.64 Å. Two additional rounds of two-dimensional classification in cryoSPARC further reduced the number of particles to 262,884. The selected particles were then subjected to 3D refinement (with the initial model created in cryoSPARC during the preliminary dataset processing) and again re-imported into RELION. Two rounds of 3D classification with local angular searches were then used to clean the particle stack. Three-dimensional refinement with an imposed C4 symmetry of 132,301 selected particles resulted in a 3.32 Å reconstruction. Refined particles were subjected to *Bayesian polishing* procedure (with un-binning to a pixel size of 0.82 Å), followed by CTF refinements, which improved resolution to 2.95 Å. After a second round of *Bayesian polishing*, particles were filtered with a third round of 3D classification. A total of 110,176 selected particles were imported into cryoSPARC for a final round of 3D refinement with imposed C4 symmetry, which resulted in a 2.65 Å reconstruction. This final map was sharpened locally with the *Local Filtering* tool in cryoSPARC with a B factor of −87 Å$^2$.

For focused MHR1/2 refinement, the final set of particles was re-imported into RELION and subjected to *C4 Symmetry Expansion* procedure, which increased the number of aligned particles to 440,704. To facilitate the local alignment of a single MHR1/2 region, the signal from all other parts of the TRPM8 reconstruction (including three additional copies of MHR1/2) was removed from the images with a *Signal Subtraction* tool. After one additional round of 3D classification, 212,851 selected particles were imported into cryoSPARC for a final round of *Local Refinement*, which resulted in a 3.20 Å reconstruction. This focused map was sharpened locally with the *Local Filtering* tool in cryoSPARC with a B factor of −148 Å$^2$.

The composite map was created in UCSF Chimera v1.17[34] by combining the global map and four copies of focused maps with the *vop maximum* command. All reported resolutions were estimated from gold-standard masked Fourier shell correlation (FSC) curves at the 0.143 threshold. Data collection and processing statistics are presented in Table 1.

**Cryo-EM map interpretation and model building**. The composite map of HsTRPM8 was used for de novo model building in Coot v0.9.8.1[35] with the aid of the orthologous structures PmTRPM8[10] (PDB ID: 6O6A/6O72) and MmTRPM8[13] (PDB ID: 7WRA) and the AlphaFold v2.1.0[36] model for the N-terminal part of MHR1/2. Multiple rounds of real-space refinement in Phenix v1.20.1-4694[37] and model building in Coot resulted in an almost complete structure of HsTRPM8 that comprised residues 43–1104, with several loops and disordered regions missing (residues 229–234, 534–556, 717–721, 914–951, 985–989, 1031–1048).

Local resolution of the HsTRPM8 map was the highest for the central part of the molecule (i.e., the pore and MHR3/4). Local resolution was the lowest for extracellular loops and C-terminal coiled-coil elements (Supplementary Fig. 2a, b). Map quality was sufficient to build most side chains in well-structured pore regions and unambiguously trace the protein backbone and some of the side chains in the lower-resolution MHR1/2 (Supplementary Fig. 2c, d). In agreement with previous reports[10], we observed additional densities in the transmembrane region that we modeled as two CHS molecules and one POPC as the most common phospholipid in Sf9 cells infected by recombinant baculovirus[38] (Fig. 3d; Supplementary Fig. 2e). We note however that due to the low local resolution of the head of this lipid we cannot exclude that this is a different phospholipid (e.g., phosphatidylethanolamine), or a mixture. Additionally, in the center of the pore, near the extracellular part, we observed clear densities that we modeled as two Na$^+$ ions (the cation at the highest concentration in the sample) and a single undecane molecule[10,13] (presumably part of an unidentified phospholipid tail; Fig. 3c).

For the preparation of Supplementary Movie 1, 3D *Variability* analysis was performed in cryoSPARC and selected components were reconstructed into three frames with *3D Variability Display* tool. Next, HsTRPM8 atomic model was fitted into these reconstructions by using a molecular dynamics flexible fitting (MDFF)[39] simulation with explicit solvent in VMD-1.9.3[40] and NAMD v2.12[41]. Morphs between these conformations were created in UCSF Chimera v1.17[34].

All figures were prepared in UCSF ChimeraX v1.6[42]. Pore radius depicted in Fig. 2a was calculated using the HOLE release 2.2.005 program[43]. Refinement and validation statistics are present in Table 1.

**Structure-based multiple sequence alignment**. Structure-based multiple sequence alignment of selected TRPM channels (Supplementary Fig. 4a) was prepared as described before[19]. Briefly, transmembrane domains of new MmTRPM8 and HsTRPM8 structures were aligned to the common reference (NvTRPM2, PDB ID: 6CO7) by using Fr-TM-Align v1.0[44] and these pairwise alignments were added to the already published multiple sequence alignment. All insertions relative to the reference 6CO7 structure were omitted from the alignment for clarity.

**Modeling of icilin binding**. The icilin complexes were generated by superimposing the HsTRPM8 structure with the avian FaTRPM8 6NR3 structure. Superimposition was guided by backbone atoms of the sensor module only (S1-S4 helices) and was performed using the VEGA ZZ 3.2.3 suite of programs[45]. By considering the different arrangements as detected for icilin in the resolved structures, an alternative structure was generated by rotating 180° the previously determined icilin poses. These obtained complexes were then minimized for 10,000 steps using NAMD[41] with the conjugated gradient algorithm, the force field CHARMM22, and a cutoff of 8 Å. All atoms outside an 8 Å radius

sphere around the icilin together with, all backbone atoms were kept fixed to preserve the experimental folding. We analyzed the HsTRPM8 structure using GeneoNet[41], a machine-learning method for pocket detection integrated into the EXSCALATE platform, and, gratifyingly, the best four detected pockets correspond to the icilin-binding site of the four protomers which further supports our model.

**HsTRPM8 activation assay.** To determine the effect of specific point mutations on TRPM8 agonist putative binding site, a response to four different agonists was tested for a set of variants of HsTRPM8 protein with selected residues in the putative ligand-binding pocket (Y745, I746, N799, and D802) mutated to alanine. To this end, HEK293 cells were transiently transfected with c-Myc-tagged TRPM8 constructs and challenged with increasing concentrations of each reference compound (icilin, menthol, Cooling Agent-10, and WS-3). All transient transfections were performed with Lipofectamine 2000 according to the manufacturer's protocol. Response was monitored using a $Ca^{2+}$ mobilization-dependent dye, Fluo-8 NW. To this end, cells were transfected and seeded 17,500 c/w in Poly-D- Lysine coated 384 MTP. 24 h after seeding, the plates were incubated for 30 min at room temperature, then the medium was manually removed and the cells were loaded with 20 μl/well of the Screen Quest$^{TM}$ Fluo-8 No Wash Calcium Assay Kit solution (0.5×). Dye-loaded cell plates were incubated for 1 h at room temperature and test compounds at 3×-concentration in 1.5% DMSO Assay Buffer were added to the wells of an Assay Plate, in 10 μl volume. The kinetic response was monitored by the FLIPR$^{TETRA}$ instrument over a period of 300 s. Finally, a second injection of 10 μl/well of reference agonists Cooling Agent-10 or Icilin at 4x-concentration in Assay Buffer was performed and the signal of the emitted fluorescence was recorded for additional 180 s. FLIPR$^{TETRA}$ settings: filters: Excitation: 470–495; Emission: 515–575; read settings (typical): Gain 130; Exp. Time 0.50; Exc.Intensity 100. The fluorescence signal was normalized to the cellular responses prior to agonist stimulation ΔF/F$_0$ (normalized MAX-MIN), where ΔF is the MAX between time points 117–182 minus the MIN between time points TP114 and TP116, and F$_0$ is the basal signal at time point TP0.

**Statistics and reproducibility.** HsTRPM8 activation assay was performed in triplicate (WS-3) or tetraplicate (icilin, menthol, Cooling Agent-10). Dose-response curves were calculated with GraphPad Prism v 6.0.7 using the log(dose) response curve with variable slope, and EC$_{50}$ values with 95% Confidence Intervals were estimated for those experiments where fit was unambiguous. Results are reported with SD error bars and with individual data points along with the fitted dose-response curves (Fig. 3i, Supplementary Fig. 5).

**Reporting summary.** Further information on research design is available in the Nature Portfolio Reporting Summary linked to this article.

## Data availability

The atomic model of HsTRPM8 is available in the Protein Data Bank (PDB) under the accession code 8BDC. The corresponding cryo-EM reconstructions are available in the EM Data Bank under the accession codes EMD-15981 (composite map), EMD-15982 (consensus map), and EMD-15983 (focused pre-MHR + MHR1/2 map). The models containing icilin molecules and source data for activation assay, multiple sequence alignment, and 3D variability results have been uploaded to public repository on the Zenodo website (https://doi.org/10.5281/zenodo.8307982).

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

## Acknowledgements

We thank Michael Arends for proofreading the manuscript. This work was financed by the statutory funding of the International Institute of Molecular and Cell Biology in Warsaw. This work was supported by the Italian Ministry of Economic Development Grant No. F/090033/01-03-04/X36 - QRARE (X36) project. This publication was developed under the provision of the Polish Ministry of Education and Science project, "Support for research and development with the use of research infrastructure of the National Synchrotron Radiation Centre SOLARIS," under contract no. 1/SOL/2021/2. We acknowledge the SOLARIS Centre for access to the cryo-EM Beamline, where the measurements were performed.

## Author contributions

S.P. established the protocol for HsTRPM8 purification and purified the protein. M.C.C. vitrified the samples and analyzed cryo-EM data. E.N. and M.C.C. built the atomic protein model. S.P., M.C.C., and M.N. wrote the manuscript. C.T., A.R.B., and G.V. reviewed the manuscript. S.G. and G.V. carried out molecular modeling studies. C.T. and G.V. reviewed the modeling results.

## Competing interests

The authors declare no competing interests. Supplementary Information is available for this paper. Correspondence and requests for materials should be addressed to Carmine Talarico or Marcin Nowotny.
