## [Peer Review File · Communications Biology]

REVIEWERS' COMMENTS:

Reviewer #1 (Remarks to the Author):

The authors satisfactorily addressed all my previous concerns and provided a sound body of additional functional data.

Therefore, I recommend publication of the manuscript.

Reviewer #2 (Remarks to the Author):

The revised manuscript titled "Structure of the Human TRPM8 Channel" by Palchevskiy et al. offers a comprehensive analysis of the structure of the human TRPM8 channel in comparison to previously published TRPM8 structures. The authors conducted molecular dynamics simulations and successfully predicted the binding site for the TRPM8 agonist icilin. In the course of the revision, the authors further enhanced the validity of their computational findings through mutagenesis and functional recording experiments, resulting in significant improvements to the manuscript.

To validate the predicted icilin binding site, the authors introduced four mutations (Y745A, I746A, N799A, and D802A). Among these mutations, Y745A can be interpreted either as a nonfunctional mutant or as a critical residue for the binding of all tested agonists. Conversely, I746A exhibited minimal impact. To bolster the robustness of the functional data, it is recommended to explore additional mutant variations, such as the L778A mutation (as suggested by the observations in Figure 3). Moreover, the usage of ligands that target different binding sites in comparison to icilin, menthol, cooling agent 10, and WS-3 would further strengthen the functional analysis. An alternative approach could involve investigating competitive/non-competitive binding through techniques like Schild regression.

Minor/Optional Comments:

- 1) Please compare the obtained EC50 values presented in the manuscript with those from published sources, whenever such information is available.
- 2) Ensure that a label is provided for N799 in Figure 3.
- 3) It appears that I746 might have been inadvertently labeled as I846 in Figure 3E-H. Please correct this label.
- 4) In the text, modify "Colling agent 10 to Cooling agent 10" to "Cooling agent 10."
- 5) The PDB validation report for HsTRPM8 is missing, which prevents the assessment of the quality of the PDB model.

In conclusion, in my opinion, the revisions have greatly enhanced the manuscript, offering valuable insights into the structure and function of the human TRPM8 channel.

Reviewer #1 (Remarks to the Author):

The authors satisfactorily addressed all my previous concerns and provided a sound body of additional functional data.

Therefore, I recommend publication of the manuscript.

We are glad to hear that the referee's assessment is positive.

Reviewer #2 (Remarks to the Author):

The revised manuscript titled "Structure of the Human TRPM8 Channel" by Palchevskiy et al. offers a comprehensive analysis of the structure of the human TRPM8 channel in comparison to previously published TRPM8 structures. The authors conducted molecular dynamics simulations and successfully predicted the binding site for the TRPM8 agonist icilin. In the course of the revision, the authors further enhanced the validity of their computational findings through mutagenesis and functional recording experiments, resulting in significant improvements to the manuscript.

To validate the predicted the icilin binding site, the authors introduced four mutations (Y745A, I746A, N799A, and D802A). Among these mutations, Y745A can be interpreted either as a nonfunctional mutant or as a critical residue for the binding of all tested agonists. Conversely, I746A exhibited minimal impact. To bolster the robustness of the functional data, it is recommended to explore additional mutant variations, such as the L778A mutation (as suggested by the observations in Figure 3). Moreover, the usage of ligands that target different binding sites in comparison to icilin, menthol, cooling agent 10, and WS-3 would further strengthen the functional analysis. An alternative approach could involve investigating competitive/non-competitive binding through techniques like Schild regression.

We agree with the referee that investigating additional mutations could further strengthen the functional analysis. Based on our structure we can predict that L778A mutation should have strong effect on the icilin binding. Other residues that could potentially be tested are E782, F839, R842, and Y1005. Testing ligands with different binding sites may also give additional insights into the binding pockets of HsTRPM8. However, we believe that these additional analysis are not necessary to support the structure and ideas presented in our manuscript and could be performed during additional functional studies of HsTRPM8 protein.

Minor/Optional Comments:

1) Please compare the obtained EC50 values presented in the manuscript with those from published sources, whenever such information is available.

We have compared our EC50 values to the literature and added the following sentences to the Results section: "For wild-type protein, the EC50 values for icilin, Cooling Agent-10, and WS-3 were similar to the values reported in previous studies (Behrendt et al., 2004). EC50 values reported for menthol varied significantly between studies (Andersson et al., 2004; Behrendt et al., 2004) and value obtained in our assay lies in the same range."

2) Ensure that a label is provided for N799 in Figure 3.

3) It appears that I746 might have been inadvertently labeled as I846 in Figure 3E-H. Please correct this label.

We have added appropriate labels for N799 and I746.

4) In the text, modify "Colling agent 10 to Cooling agent 10" to "Cooling agent 10."

We have modified the name accordingly.

5) The PDB validation report for HsTRPM8 is missing, which prevents the assessment of the quality of the PDB model.

The missing validation report is now uploaded to the system

In conclusion, in my opinion, the revisions have greatly enhanced the manuscript, offering valuable insights into the structure and function of the human TRPM8 channel.

We are glad to hear that.

In addition, we have introduced the following editorial changes:

1. Edits of the figures.

All figures and supplementary figures were adjusted to meet the guidelines for the final submission.

Additional changes:

- Figure 2: insets were labeled 'd' and 'e'.
- Figure 3: 'Icillin' was corrected to 'Icilin'.
- Figure 3: Individual datapoints were added to the plots as requested.
- Supplementary Figure 1a: scale bar was added to the micrograph.

2. Edits in the text.

The manuscript was divided into Main manuscript file (Word document) and Supplementary Information (pdf file) and all edits were introduced as requested in the editorial requests file.

Supplementary Tables S1 and S2 were merged into Table 1 provided at the end of document. We note that some values in Table 1 (i.e., number of atoms and residues, R.m.s. deviations, and Ramachandran plot statistics) are updated – these statistics are either in the same range or slightly better than the previous ones.

Additional small editorial changes:

- 'modelling' was corrected to 'modeling' in all places
- Page 5: 'D802 variant showed significantly reduced response' changed to 'D802 variant showed much weaker response.' as requested
- Figure 3i caption was updated
- Reference to NAMD added to the Materials and Methods section
- *Statistics and reproducibility* section was added; relevant information from *HsTRPM8 activation assay* section was edited and moved there.
- *Data availability* section was updated with link to Zenodo repository